# High-Accuracy Phase Frequency Detection Technology Based on BDS Time and Frequency Signals

**DOI:** 10.3390/s24144606

**Published:** 2024-07-16

**Authors:** Baoqiang Du, Lanqin Tan

**Affiliations:** School of Information Science and Engineering, Hunan Normal University, Changsha 410081, China; tanlq@hunnu.edu.cn

**Keywords:** BDS, phase measurement, frequency measurement, greatest common factor period

## Abstract

For the time and frequency signals of Beidou satellites, a high-accuracy phase frequency detection technology based on phase group synchronization is proposed. Using the Beidou receiver and satellite signals as the frequency standard and the measured signals, respectively. The Beidou receiver and the satellite signals are sent to the phase coincidence detector of the different frequencies to generate a phase coincidence point pulse, which is sent to the different frequency phase detector as a control signal to generate the phase differences between the Beidou receiver and satellite signals, and then complete the high-accuracy phase synchronization between the Beidou receiver and satellite signals. Experimental results show that when the delay resolution reaches ps level, the phase synchronization accuracy of the system can reach 10 ps, which has the characteristics of small phase noise, low development cost, simple circuit structure, and high synchronization accuracy compared with the traditional phase synchronization technologies. Therefore, it would be widely used in satellite positioning, astrometry, precision navigation, aerospace, satellite launch, power transmission, communications, radar, and other high-tech fields.

## 1. Introduction

Different frequency phase synchronization refers to the process by which the phase differences between different frequency signals are cleared or processed with minimization, and the processed phase differences do not change over time. High-precision phase synchronization is based on high-resolution phase difference measurements. Phase difference is an important parameter in the control and measurement of the BDS (Beidou Navigation Satellite System) time and frequency signals, which is widely used in power system time synchronization, aerospace, satellite navigation, communications, radar, astronomy, and other fields and has an important influence on the whole performance of the system. Therefore, high-precision phase difference measurement has gradually become an important research key in the Beidou time and frequency measuring and control fields [1,2,3,4]. However, the measurement of the phase difference is different from the traditional measurement of voltage or current signals. First, the phase difference signal is attached to the voltage or current signal, so how to eliminate the effects of frequency variation and phase noise is very important for phase difference measurement. Second, the phase difference is a comparative quantity, and when measuring, it is not only necessary to make the frequency of the two signals the same but also to exclude the influence of factors such as the inconsistency of the amplitude of the two signals.

At present, the commonly used phase difference measurement methods mainly include the Lissajous figures method [5,6], the numerical sampling method, the phase-voltage conversion method, and the phase-time conversion method, etc. The Lissajous figures method is mainly used to obtain the phase difference by observing the Lissajous figures method on the oscilloscope. This method is relatively simple, but because it is based on direct observation, it is difficult to guarantee its measurement accuracy, and cannot guarantee measurement automation. The numerical sampling method mainly uses synchronous sampling technology to obtain the sampling value of the two input signals and obtains the phase angle by processing the instantaneous amplitude [7,8,9]. This method may get quite accurate results for low frequencies, but when the measured frequency is relatively high, the system requires too high processing speed for the MCU (Microcontroller Unit) due to the high sampling frequency and the strict synchronization of sampling. If the processing speed of the MCU is not enough, the probability of error will be greatly increased. The phase-voltage conversion method is mainly measured by converting the phase signal into the voltage signal, and the structure is relatively simple [10,11]. However, due to the nonlinear distortion of the low-pass filter, the ADC (Analog-to-Digital Converter) will introduce quantization errors, so its measurement accuracy is often not ideal. The phase-time conversion method is mainly to shape the measured signal and the signal after the phase shift, respectively, and then get the phase difference by counting. However, due to the counting error of ±1 word in the common counting method, the measuring error is large, and further improving the measuring accuracy is difficult.

Aiming at the defects of the traditional phase synchronization detection technologies, a high-accuracy phase frequency detection technology based on the time and frequency signals of Beidou satellites is proposed in this paper. Based on the periodic variation rule of group phase, this technology introduces the different frequencies phase coincidences detection’s principle to accurately capture the phase coincidence between the Beidou receiver and satellite signals, eliminating the ±1-word counting error in the traditional phase synchronization detection and directly completing the phase synchronization detection without frequency normalization processing [12,13,14,15]. The proposed method not only has wide application in the precision measurement and control of frequency and phase in the high-accuracy time-frequency transmission of Beidou satellite, as well as the signal processing and time synchronization of space-borne atomic clock in the Beidou navigation system, but also reduces the development cost, optimizes the circuit structure, and also improves the resolution of phase synchronization detection of the Beidou navigation satellite.

## 2. Different Frequency Phase Synchronization Principle

In Beidou signal transmission, due to the influence of external interference and system noise, the frequency relations between the Beidou satellite and receiver signals are not fixed but have a certain relative frequency difference, leading to the frequency relations’ complexity [16,17]. In the phase synchronization detection of the Beidou satellite and receiver signals, the phase comparison relations between them are mainly manifested as phase group synchronization, group period, and the greatest common factor period.

Assuming that the Beidou receiver signal is $f_{R}=Af_{c}$ and its period is $T_{R}=mT_{c}$, the Beidou satellite signal is $f_{s}=Bf_{c}$ and its period is $T_{s}=nT_{c}$, where the A,B,m,n are positive integers of mutual prime, then
(1)$$(f_{R},f_{s})=f_{c}=\frac{1}{T_{m}}$$
where the $f_{\max c}$ and the $T_{m}$ are the greatest common factor frequency and the least common multiple period, respectively.

In addition, the $T_{m}$ can also be obtained directly by the Beidou receiver and satellite signals.
(2)$$T_{m}=mnT_{c}$$

Here $T_{c}$ is called the greatest common factor period.
(3)$$T_{c}=(T_{R},T_{s})$$

According to (1) and (3),
(4)$$\frac{1}{f_{c}}=mnT_{c}$$

Let
(5)$$f_{q}=mnf_{c}$$

Then
(6)$$f_{q}=\frac{1}{T_{c}}$$

The $f_{q}$ is known as the phase quantization frequency, and the $T_{c}$ is also known as the quantization phase resolution or phase quantization.

According to (4), $f_{c}=\frac{1}{mnT_{c}}=\frac{1}{nT_{R}}=\frac{T_{c}}{nT_{c}T_{R}}=\frac{T_{c}}{T_{R}T_{s}}=T_{c}f_{R}f_{s}$, so $T_{c}=\frac{f_{c}}{f_{R}f_{s}}=\frac{f_{c}}{Af_{c}\cdot Bf_{c}}=\frac{1}{ABf_{c}}$.

In combination (6),
(7)$$f_{q}=ABf_{c}$$

Since $f_{R}=Af_{c}$ and $T_{R}=mT_{c}$, then $\frac{1}{f_{R}}=\frac{1}{Af_{c}}=T_{R}=mT_{c}$ and $\frac{1}{Am}=f_{c}T_{c}=\frac{T_{c}}{T_{m}}=\frac{T_{c}}{mnT_{c}}=\frac{1}{mn}$. So
(8)A=n

Since $f_{s}=Bf_{c}$ and $T_{s}=nT_{c}$, then $\frac{1}{f_{s}}=\frac{1}{Bf_{c}}=T_{s}=nT_{c}$ and $\frac{1}{Bn}=f_{c}T_{c}=\frac{T_{c}}{T_{m}}=\frac{T_{c}}{mnT_{c}}=\frac{1}{mn}$. So
(9)B=m

If the Beidou receiver signal is divided by $f_{R}=13$ MHz as the frequency standard signal, and the Beidou satellite signal is divided by $f_{s}=21$ MHz as the measured signal, their maximum common factor frequency is 1 MHz, and their minimum common multiple period is 1000 ns, as shown in Figure 1.

From Figure 1, it can be seen that two signals $f_{R}$ and $f_{s}$ with different frequencies do not have phase continuity with their respective periods as time intervals. Therefore, in the common phase processing technologies, the different frequencies’ signals cannot be compared and must be normalized. However, two signals $f_{R}$ and $f_{s}$ with different frequencies have phase synchronization or phase continuity with the $T_{c}$ intervals, which means that the $T_{c}$ connects two signals $f_{R}$ and $f_{s}$ of different frequencies through phase.

According to (2),
(10)$$T_{m}=mT_{s}=nT_{R}$$

In every $T_{m}$, the average of all the phase differences is equal. If all the phase differences in a $T_{m}$ are regarded as a phase difference whole, it is called a group. Every phase difference in the $T_{m}$ maintains strict phase synchronization at group intervals, which is called phase group synchronization.

During the different-frequency phase comparison, the phase difference at the $T_{m}$ is the fixed initial phase difference between the $f_{R}$ and $f_{s}$ signals. If the fixed phase difference is zero, the magnitude of other phase differences or the variation of other phase differences within the $T_{m}$ is the integer multiple of the $T_{c}$ or the quantization phase resolution, that is, in a $T_{m}$, the $f_{R}$ is used as the frequency standard signal, for the $f_{R}$ signal’s each rising edge, the phase difference between the adjacent $f_{s}$ signal’s rising edge is the direct phase comparison result of the two signals $f_{out}$, that is, ${\left[PD_{1},PD_{2},PD_{3},\;\cdots ,PD_{B}\right]}^{T}$,
(11)$$f_{out}=\left[\begin{array}{l} PD_{1}\\ PD_{2}\\ \;\vdots \\ PD_{B} \end{array}\right]=\left[\begin{array}{l} n_{1}T_{s}-1T_{R}\\ n_{2}T_{s}-2T_{R}\\ \;\;\;\;\vdots \\ AT_{s}-BT_{R} \end{array}\right]=\left[\begin{array}{l} n_{1}BT_{c}-1AT_{c}\\ n_{2}BT_{c}-2AT_{c}\\ \;\;\;\;\;\;\;\vdots \\ ABT_{c}-BAT_{c} \end{array}\right]\;=\left[\begin{array}{l} n_{1}B-1A\\ n_{2}B-2A\\ \;\;\;\;\;\vdots \\ AB-BA \end{array}\right]T_{c}=\left[\begin{array}{l} g_{1}\\ g_{2}\\ \vdots \\ g_{B} \end{array}\right]T_{c}$$
where $n_{1},n_{2},n_{3},\cdots ,A$ is the number of periods of each rising edge signal $f_{R}$ for signal $f_{s}$, and ${\left(g_{1},g_{2},g_{3},\cdots ,g_{B}\right)}^{T}$ is a positive integer. From (11), it can be seen that $f_{out}$ is an integral multiple of the $T_{c}$, that is, each phase difference in the group is quantized by the $T_{c}$. The greatest common factor period, as the most basic unit to quantify the phase comparison’s result, is called phase quantization, which is determined by the frequency relationship between the signals $f_{R}$ and $f_{s}$.

From (11), it can be seen that the smallest phase difference in a $T_{m}$ is the quantization phase resolution, also known as the measurement resolution, which is usually greater than the phase detecting resolution of the detection device [18]. The phase detection resolution is determined by the phase detector, and different phase detectors have different phase detection resolutions, which are mainly determined by the structure and composition of the phase detector. The phase synchronization detector is composed of a TTL (Transistor-Transistor Logic) logic circuit. Due to its switching speed, the phase detection resolution can generally only reach 2 ns; that is, in a $T_{m}$, all the phase differences less than 2 ns will not be identified or detected, and then the phase detection circuit will output almost identical results for each phase difference that cannot be detected, which is called the fuzzy area. The fuzzy area is actually a lot of phase differences that cannot be distinguished by phase detection resolution, that is, a cluster of phase differences in which the phase difference cannot be detected by the phase detector. The phase differences in the clusters are actually the phase coincidence points between the signals $f_{R}$ and $f_{s}$, but each phase coincidence point represents a different phase coincidence degree between the signals $f_{R}$ and $f_{s}$. As shown in Figure 2.

As can be seen from Figure 2, the fuzzy area consists of phase differences with different coincidence degrees and cannot be identified by phase detection resolution, and its envelope presents a normal distribution, which is mainly caused by the phase advance, strict coincidence, and hysteresis of signal $f_{s}$ to $f_{R}$. If the frequency relations between the signals $f_{R}$ and $f_{s}$ is fixed, the fuzzy area is stable and synchronized with the $T_{m}$ intervals. The phase differences of the fuzzy area edge are very close to the phase detection resolution of the phase detector, which will be affected by external interference, system noise, and other factors that delay the phase difference. The phase difference characteristic pulse of the fuzzy area edge, referred to as the fuzzy area characteristic pulse, has a small range of jitter, that is, the instantaneous recognition of the fuzzy area characteristic pulse by phase detection, also known as the systematic error of the phase detector. Because the phase detecting resolution is fixed, the characteristic pulse of the fuzzy area has high stability.

## 3. Different Frequency Phase Synchronization Scheme

The detection scheme for different frequency phases of synchronization is shown in Figure 3. First, the Beidou receiver and satellite signals are adjusted to make them narrow pulses suitable for phase synchronous detection. Second, the two narrow pulses are detected by different frequency phase comparators and phase coincidence detectors, respectively. The different frequency phase comparator outputs the phase difference pulse signal, and the phase coincidence detector outputs the fuzzy area pulse signal. Third, the fuzzy area pulse signal is transmitted to the phase synchronization detector as a control signal to generate the counter’s switching signal, that is, the gate time interval. Finally, the two narrow pulse signals are counted in the time interval of the gate, and the Beidou satellite signal’s frequency is acquired through data processing. The Beidou satellite signal’s frequency measurement precision reflects the accuracy of different frequency phase synchronizations.

### 3.1. Signal Conditioning Circuit

It is inevitable that Beidou satellite signals will be affected by noise and interference (such as different kinds and intensities of electromagnetic wave interference on Earth, mutual interference between satellites, shielding environment interference, closed space interference, natural factors interference, man-made interference, equipment fault interference, cross-polarization interference, and so on) during transmission. To reduce the false triggering of phase coincidence detection caused by excessive phase distortion caused by noise and interference during the transmission of analog electrical signals and to improve the measurement accuracy of the phase synchronization detection circuit, it is necessary to condition and shape the Beidou receiver and satellite signals to become the same frequency’s narrow pulse signal. The same frequency pulse width of the Beidou receiver and satellite signals determines the measurement accuracy of the phase synchronization detector, so to obtain a suitable pulse width of the same frequency, a differential adjustable delay technology is adopted [19], as shown in Figure 4.

In Figure 4, the delay unit is composed of fibers of different lengths, and generally, the length of delay 1 is slightly greater than that of delay 2. When the length difference between delay 1 and delay 2 is 3 mm, the delay resolution can reach 10 ps, and the same frequency pulse width can also reach 10 ps. The same-frequency pulse’s experiment results are shown in Figure 5.

### 3.2. Different Phase Comparison Circuit

The phase difference between the Beidou satellite and receiver signals, that is, the result of phase synchronization between the satellite and the ground, can be obtained directly by the different frequency phase comparator. To ensure the phase detection output’s stability, the phase difference measurement circuit here uses an R-S flip-flop as a phase comparator, as shown in Figure 6.

When the pulse signal of the Beidou receiver is a high-level input and the Beidou satellite pulse signal is a low-level input R-S trigger, the R-S trigger output is high level. When the Beidou receiver pulse signal and the Beidou satellite pulse signal are low-level input, the R-S trigger maintains a high-level output. When the pulse signal of the Beidou receiver is a low-level input and the Beidou satellite pulse signal is a high-level input R-S flip-flop, the R-S trigger output is low-level. In this way, the phase difference measurement results between the Beidou receiver and satellite signals in the $T_{m}$ can be obtained. If the Beidou receiver signal $f_{R}=4.012\;\mathrm{MHz}$ and the Beidou satellite signals $f_{s}=5\;\mathrm{MHz}$, the phase difference measuring results $f_{out}$ between them are shown in Figure 7.

### 3.3. Different Frequency Phase Coincidence Circuit

When the Beidou receiver pulse signal and satellite pulse signal are input R-S flip-flops at high levels at the same time, that is, when the Beidou receiver pulse signal and satellite pulse signal have phase coincidence, the R-S flip-flop’s output is in an uncertain state, and the triggering of the counter gate has great randomness. To eliminate the randomness of the counter gate trigger, a phase coincidence detection circuit based on a common frequency source is used, which is composed of an edge-type D flip-flop. In this circuit, the fuzzy area’s edge characteristic pulse is generated. Furthermore, necessarily reducing the fuzzy area width, the trigger pulse of the phase coincidence fuzzy area is extracted effectively to ensure the trigger stability of the counter gate. To this end, a binary data selector is used here, and when the effective phase of the Beidou receiver pulse signal coincides with the Beidou satellite pulse signal, the data selector outputs a high-level pulse as the switching signal of the counter gate. As shown in Figure 8.

If the Beidou receiver signal $f_{R}=10.23\;\mathrm{MHz}$ and the Beidou satellite signal $f_{s}=8\;\mathrm{MHz}$, the phase synchronization result, that is, the counter gate switch pulse signal, is shown in Figure 9.

### 3.4. Data Processing and Display

When the frequency deviation Δf≠0 between the Beidou receiver signal and satellite signal, if the Beidou receiver signal’s frequency $f_{R}^{\prime }=f_{R}\pm \Delta f$,Δf>0 and the period $T_{R}^{\prime }=T_{R}\mp \Delta t$, Δt>0, then the Beidou satellite signal’s frequency $f_{s}^{\prime }=f_{s}$ and the period $T_{s}^{\prime }=T_{s}$ Where Δt is a time drift quantity caused by the Δf. The conditions of all the phase differences in a $T_{m}$ is as follows:

The phase difference’s change within the first $T_{m}$ is shown in Formula (12).
(12)$$\left[\begin{array}{l} PD_{1}^{\prime }\\ PD_{2}^{\prime }\\ PD_{3}^{\prime }\\ \;\vdots \\ PD_{B}^{\prime } \end{array}\right]=\left[\begin{array}{l} n_{1}T_{s}-1T_{R}^{\prime }\\ n_{2}T_{s}-2T_{R}^{\prime }\\ n_{3}T_{s}-3T_{R}^{\prime }\\ \;\;\;\;\vdots \\ AT_{s}-BT_{R}^{\prime } \end{array}\right]=\left[\begin{array}{l} n_{1}T_{s}-(T_{R}\mp \Delta t)\\ n_{2}T_{s}-2(T_{R}\mp \Delta t)\\ n_{3}T_{s}-3(T_{R}\mp \Delta t)\\ \;\;\;\;\vdots \\ AT_{s}-B(T_{R}\mp \Delta t) \end{array}\right]=\left[\begin{array}{l} n_{1}T_{s}-T_{R}\pm \Delta t\\ n_{2}T_{s}-2T_{R}\pm 2\Delta t\\ n_{3}T_{s}-3T_{R}\pm 3\Delta t\\ \;\;\;\;\vdots \\ \;\;\;\pm B\Delta t \end{array}\right]$$

The phase difference’s change within the second $T_{m}$ is shown in Formula (13).
(13)$$\left[\begin{array}{l} PD_{1}^{\prime \prime }\\ PD_{2}^{\prime \prime }\\ PD_{3}^{\prime \prime }\\ \;\vdots \\ PD_{B}^{\prime \prime } \end{array}\right]=\left[\begin{array}{l} (A+n_{1})T_{s}-(1+B)T_{R}^{\prime }\\ (A+n_{2})T_{s}-(2+B)T_{R}^{\prime }\\ (A+n_{3})T_{s}-(3+B)T_{R}^{\prime }\\ \;\;\;\;\vdots \\ 2AT_{s}-2BT_{R}^{\prime } \end{array}\right]=\left[\begin{array}{l} \left(A+n_{1})T_{s}-(1+B)(T_{R}\mp \Delta t\right)\\ \left(A+n_{2})T_{s}-(2+B)(T_{R}\mp \Delta t\right)\\ \left(A+n_{3})T_{s}-(3+B)(T_{R}\mp \Delta t\right)\\ \;\;\;\;\vdots \\ 2AT_{s}-2B(T_{R}\mp \Delta t) \end{array}\right]\;=\left[\begin{array}{l} n_{1}T_{s}-T_{R}\pm \Delta t\pm B\Delta t\\ n_{2}T_{s}-2T_{R}\pm 2\Delta t\pm B\Delta t\\ n_{3}T_{s}-3T_{R}\pm 3\Delta t\pm B\Delta t\\ \;\;\;\;\;\;\;\;\;\vdots \\ \;\;\;\;\;\;\;\;\pm 2B\Delta t \end{array}\right]$$

The phase difference’s change within the third $T_{m}$ is shown in Formula (14).
(14)$$\begin{array}{l} \left[\begin{array}{l} PD_{1}^{\prime \prime \prime }\\ PD_{2}^{\prime \prime \prime }\\ PD_{3}^{\prime \prime \prime }\\ \;\;\vdots \\ PD_{B}^{\prime \prime \prime } \end{array}\right]=\left[\begin{array}{l} (2A+n_{1})T_{s}-(1+2B)T_{R}^{\prime }\\ (2A+n_{2})T_{s}-(2+2B)T_{R}^{\prime }\\ (2A+n_{3})T_{s}-(3+2B)T_{R}^{\prime }\\ \;\;\;\;\vdots \\ 3AT_{s}-3BT_{R}^{\prime } \end{array}\right]=\left[\begin{array}{l} \left(2A+n_{1})T_{s}-(1+2B)(T_{R}\mp \Delta t\right)\\ \left(2A+n_{2})T_{s}-(2+2B)(T_{R}\mp \Delta t\right)\\ \left(2A+n_{3})T_{s}-(3+2B)(T_{R}\mp \Delta t\right)\\ \;\;\;\;\vdots \\ 3AT_{s}-3B(T_{R}\mp \Delta t) \end{array}\right]\\ =\left[\begin{array}{l} n_{1}T_{s}-T_{R}\pm \Delta t\pm 2B\Delta t\\ n_{2}T_{s}-2T_{R}\pm 2\Delta t\pm 2B\Delta t\\ n_{3}T_{s}-3T_{R}\pm 3\Delta t\pm 2B\Delta t\\ \;\;\;\;\;\;\;\;\;\vdots \\ \;\;\;\;\;\;\;\pm 3B\Delta t \end{array}\right] \end{array}$$

The phase difference’s change within the (*n* − 1)th $T_{m}$ is shown in Formula (15).
(15)$$\left[\begin{array}{l} PD_{1}^{\left(n-1\right)}\\ PD_{2}^{\left(n-1\right)}\\ PD_{3}^{\left(n-1\right)}\\ \;\;\vdots \\ PD_{B}^{\left(n-1\right)} \end{array}\right]=\left[\begin{array}{l} {}[(n-2)A+n_{1}]T_{s}-[1+(n-2)B]T_{R}^{\prime }\\ {}[(n-2)A+n_{2}]T_{s}-[2+(n-2)B]T_{R}^{\prime }\\ {}[(n-2)A+n_{3}]T_{s}-[3+(n-2)B]T_{R}^{\prime }\\ \;\;\;\;\;\;\;\;\;\;\;\vdots \\ (n-1)AT_{s}-(n-1)BT_{R}^{\prime } \end{array}\right]=\left[\begin{array}{l} \left[(n-2)A+n_{1}]T_{s}-[1+(n-2)B](T_{R}\mp \Delta t\right)\\ \left[(n-2)A+n_{2}]T_{s}-[2+(n-2)B](T_{R}\mp \Delta t\right)\\ \left[(n-2)A+n_{3}]T_{s}-[3+(n-2)B](T_{R}\mp \Delta t\right)\\ \;\;\;\;\;\;\;\;\;\;\;\vdots \\ \left(n-1)AT_{s}-(n-1)B(T_{R}\mp \Delta t\right) \end{array}\right]\;=\left[\begin{array}{l} n_{1}T_{s}-T_{R}\pm \Delta t+(n\pm 2)B\Delta t\\ n_{2}T_{s}-2T_{R}\pm 2\Delta t+(n\pm 2)B\Delta t\\ n_{3}T_{s}-3T_{R}\pm 3\Delta t+(n\pm 2)B\Delta t\\ \;\;\;\;\;\;\;\;\;\vdots \\ \;\;\;\;\;\pm (n-1)B\Delta t \end{array}\right]$$

The phase difference’s change within the *n*th $T_{m}$ is shown in Formula (16).
(16)$$\begin{array}{l} \left[\begin{array}{l} PD_{1}^{\left(n\right)}\\ PD_{2}^{\left(n\right)}\\ PD_{3}^{\left(n\right)}\\ \;\vdots \\ PD_{B}^{\left(n\right)} \end{array}\right]=\left[\begin{array}{l} {}[(n-1)A+n_{1}]T_{s}-[1+(n-1)B]T_{R}^{\prime }\\ {}[(n-1)A+n_{2}]T_{s}-[2+(n-1)B]T_{R}^{\prime }\\ {}[(n-1)A+n_{3}]T_{s}-[3+(n-1)B]T_{R}^{\prime }\\ \;\;\;\;\;\;\;\;\;\;\;\vdots \\ \;\;\;\;\;\;nAT_{s}-nBT_{R}^{\prime } \end{array}\right]=\left[\begin{array}{l} \left[(n-1)A+n_{1}]T_{s}-[1+(n-1)B](T_{R}\mp \Delta t\right)\\ \left[(n-1)A+n_{2}]T_{s}-[2+(n-1)B](T_{R}\mp \Delta t\right)\\ \left[(n-1)A+n_{3}]T_{s}-[3+(n-1)B](T_{R}\mp \Delta t\right)\\ \;\;\;\;\;\;\;\;\;\;\;\vdots \\ \;\;\;\;nAT_{s}-nB(T_{R}\mp \Delta t) \end{array}\right]\\ =\left[\begin{array}{l} n_{1}T_{s}-T_{R}\pm \Delta t\pm (n-1)B\Delta t\\ n_{2}T_{s}-2T_{R}\pm 2\Delta t\pm (n-1)B\Delta t\\ n_{3}T_{s}-3T_{R}\pm 3\Delta t\pm (n-1)B\Delta t\\ \;\;\;\;\;\;\;\;\;\vdots \\ \;\;\;\;\;\;\;\pm nB\Delta t \end{array}\right] \end{array}$$

It can be seen from Formulas (12)–(16) that, in the different frequencies phase comparison, we can get the scientific change rule of group quantization and phase difference. In the contiguous $T_{m}$, any phase difference’s stepping $\Delta t=PD_{B}^{\left(n\right)}-PD_{B-1}^{\left(n\right)},\;B\geq 2$, and the group quantization’s stepping ΔP=BΔt.
(17)$$\Delta P=|PD_{B}^{\left(n\right)}-PD_{B}^{\left(n-1\right)}|,\;n\geq 2$$

When any phase difference’s stepping equals the $T_{s}$ in the different frequencies phase comparison, the group quantization at the $T_{m}$ and all the phase differences in a $T_{m}$ will undergo periodic changes, and repeatedly return to the initial state, turn round and round in turn.
(18)$$n\Delta P=T_{s}$$
$$nB\Delta t=BT_{c},\;n\Delta t=T_{c}$$
(19)$$BT_{c}=T_{s}$$

Therefore, when $T_{R}:T_{s}=A:B$, the highest measuring resolution equals the $T_{c}$, when $T_{R}:T_{s}\neq A:B$, the highest measuring resolution equals the Δt. The time experienced by a periodic change of any phase difference and group quantization is given by a group period $T_{g}$, which is written as follows,
(20)$$T_{g}=n(T_{m}\mp \Delta P)$$

Thus, the group period can be obtained by observing the two-phase differences of the repetition time, and the time drift quantity Δt can be calculated according to the phase difference’s stepping.
(21)$$\Delta t=\frac{T_{s}}{nB}=\frac{T_{s}T_{m}}{nT_{m}B}=\frac{T_{s}T_{m}}{T_{g}B}$$

The real frequency standard signal’s period $T_{R}^{\prime }$ can be acquired by Formula (21).
(22)$$T_{R}^{\prime }=T_{R}\mp \Delta t=T_{R}\mp \frac{T_{s}T_{m}}{T_{g}B}$$

From above, if using the group period as the measuring gate, the time interval between fuzzy areas is usually used as the measuring gate in practical applications and research. The Beidou satellite signal’s actual frequency can be obtained by counting the Beidou receiver and satellite signals at gate time.
(23)$$N_{R}T_{R}^{\prime }=N_{s}T_{s}^{\prime },\;f_{s}=\frac{N_{s}}{N_{R}T_{R}^{\prime }}$$
where $N_{s}$ is the Beidou satellite signal’s counting value and $N_{R}$ is the Beidou receiver signal’s counting value. These data are mainly processed by single chip microcomputer MCU, and the results of the data processing are displayed on an LCD (Liquid Crystal Display).

In a word, the different frequency phase comparison technology does not adopt the production process improvement or the optimization algorithm in the common phase comparison technology but applies the phase difference variation’s scientific rule to frequency measurement so as to solve the ±1 word counting error and improve the measurement accuracy.

## 4. Experimental Results and Their Analysis

According to the scheme shown in Figure 3, the system prototype has been developed and tested.

### 4.1. Phase Difference Measurement Experiment

In the phase difference measuring experiment, the system uses the Agilent8664 synthesizer made by Agilent Company in the Santa Clara, CA, United States to generate the Beidou receiver signal $f_{R}=10\;\mathrm{MHz}$ and the Beidou satellite signal $f_{s}=10.23\;\mathrm{MHz}$. The counter reference gate time is set to 1 s. To make full use of the phase group synchronization phenomenon naturally formed between the Beidou receiver and satellite signals and improve the phase difference’s measurement resolution, the input connection wires of the two signals are set to different lengths. The input cable of the Beidou receiver signal is 12 cm longer than the input cable of the Beidou satellite signal. This is due to the fact that the transmission speed of high-frequency signals in the connection line is close to the speed of light, the phase delay of the 1-m optical fiber connection line is 3.33 ns, and the phase difference between the Beidou receiver and satellite signals is 4 degrees. Then the phase difference produced by the 12-cm connection line can be delayed by 1.444 degrees. As shown in Figure 10.

From the experimental results of real-time phase difference measurement, the phase difference measuring technology based on phase group synchronization can achieve a measuring resolution of 10 ps, and there are only a few ps jumps between two adjacent measurements. The phase difference between the maximum and minimum values is about 16 ps.

### 4.2. Frequency Measurement Experiment

At present, an engineering prototype DF427 high-precision frequency measurement meter designed by the scheme shown in Figure 3 has been produced. The engineering prototype’s frequency measurement range is 2–150 MHz, the frequency synchronization accuracy, that is, the frequency difference, is better than 1 Hz, and the system’s frequency stability reaches 10^−14^/s. To verify the engineering prototype’s frequency accuracy, a 10 MHz signal generated by a crystal oscillator OCXO 8607-BM (OSCILIOQUARTZ, Neuchatel, Switzerland) with ultra-high stability is used as the system’s external frequency standard, and a TSG4102A frequency signal source is used to generate the measured signal. The measurement results are shown in Table 1.

In the previous research on frequency-time synchronization in the literature, we have also completed a lot of research. Phase group synchronization technology has a certain basis [5]. On the basis of phase group synchronization theory, we obtain a frequency stability of 3.7 × 10^−12^/s. According to the data shown in Table 1, for RF signals with complex frequency relationships, such as 16.384 MHz signals, the frequency stability with the 10^−13^/s level can easily be achieved, and special frequency signals, such as 10 MHz integer frequency signals, can get the 10^−14^/s level. These are mainly for the following reasons: First, phase coincidence detection technology based on a common frequency source is used to generate the fuzzy area’s edge characteristic pulse as the counter’s switching signal, which improves the counter gate’s stability and the system measurement’s phase synchronization accuracy. In the phase coincidence detection circuit, the common frequency source signal with high stability detects the group phase coincidences between the Beidou satellite and receiver signals. Because the frequency of the used common frequency source has a suitable frequency deviation from the Beidou receiver and satellite signals, a high precision of group phase coincidence detection can be achieved. The frequency measurement technology based on group phase relation is used to measure the group phase coincidence in high-resolution phase difference. The group period’s integer multiple time is used as the frequency measuring gate, so that the starting and stopping of the gate are synchronized with the two group phase coincidence points. According to the law of group phase relationship, the time interval between the phase coincidence points between the signals is exactly an integer multiple of the period of the common frequency signal, so the counting error can be greatly reduced by increasing the clock signal of the common frequency source as the counter. Here, the frequency and the common frequency source’s stability have a great influence on the measured results. The higher the measurement accuracy, the higher the requirement for the phase coincidence capture circuit. Second, phase group synchronization technology is used to process the phase quantization of different frequency phase comparison results, which reduces the requirement for a detection circuit and improves the resolution of phase measurement.

### 4.3. Comparison with the Traditional Frequency Measurement Methods

The dual mixer time difference measuring device built by American National Standards Institute Allen has played an important role in the time-domain measurement of Global Positioning System (GPS) satellite clocks.

In traditional frequency measurement technology, the dual mixer time difference measuring technology is now the most widely applied and precise frequency measuring technology internationally. On the one hand, the measured frequency is transformed into a lower beat frequency with the help of a low noise dual mixer and a common frequency source, and the time difference amplification and frequency difference multiplication are realized. On the other hand, the average time interval measuring technology is adopted, and the counter is in control through a divider, so that the quantization error is reduced statistically to improve the measurement resolution. The frequency measuring range of the dual mixer time difference measuring device is generally 1–30 MHz. If the frequency measuring range is to be widened, more mixers must be added. The noise produced by the added mixers and the frequency normalization device increases the gate switch signal’s jitter, and the system’s frequency stability decreases. Therefore, it is not very easy to achieve both a wide frequency measuring range and high frequency stability in the dual mixer time difference measurement.

Currently, the frequency stability of any frequency signal system in the RF range can be up to 10^−14^/s. For example, for 10 MHz and 5 MHz frequency signals, the frequency stability of the system can reach the order of 2.5 × 10^−14^/s and 5 × 10^−14^/s, respectively, and the frequency synchronization accuracy is 25 Hz. These are slightly less accurate in measurement than the frequency stability values in Table 1, such as 1.54 × 10^−14^/s, and less accurate in frequency synchronization than the frequency synchronization values in Table 1, such as less than 1 Hz. The DF427 high-accuracy frequency measuring meter based on Beidou time and frequency signals is the most advanced frequency and phase synchronous measurement instrument in the world. That is, except for the DF427 high-precision frequency measurement, there is no time-frequency measurement instrument with frequency precision less than 1 Hz and phase synchronization precision less than 10 ps in the world, which has been proved by the experimental data in Table 1. Because the dual mixer time difference measuring technology mainly depends on the frequency difference doubling effect to enhance the frequency stability, provided the time interval counter measuring resolution of 25 ns, the system’s frequency stability can achieve 2.5 × 10^−14^/s, but it is difficult to raise the system’s accuracy due to the counter clock frequency’s limitation and the counting error of ±1 word, and the highest frequency synchronization accuracy can only reach 25 Hz. In practical applications, the dual mixer time difference measuring device is expensive to develop, huge in volume, complex in structure, and high in market price, which severely limits its wide application.

## 5. Conclusions

A high-accuracy phase frequency detection technology based on BDS time and frequency signals is proposed. Combining the phase group synchronization theory and the different frequency phase coincidence detection technology based on a common frequency source, a Beidou high-precision different frequency phase synchronization detection scheme is designed. Using the group phase relationship, the group phase coincidences are used for phase synchronous detection between the Beidou receiver and satellite signals, which improves the phase difference’s measuring resolution and time synchronous accuracy between satellite and earth. Applying phase group synchronization to high-accuracy phase frequency measurement is a new breakthrough in Beidou time-frequency measuring fields, which gets rid of the traditional frequency measurement of the same frequency only phase comparison and frequency normalization processing but applies the phase’s variation rule between different frequency signals to the frequency measurement and can complete the phase comparison and control without the frequency normalization processing of different frequency signals. It is very difficult to achieve the 10 ps phase difference measurement resolution due to the frequency normalization and the ±1-word counting error in the traditional phase synchronization method. However, the experimental results indicate that the proposed method can achieve phase measurement resolution better than 10 ps in phase synchronization detection, frequency synchronization accuracy better than 1 Hz in frequency measurement, and second-level frequency stability of 10^−14^/s. The Beidou Navigation System is one of the four global positioning systems. It has the same functions as other satellite navigation systems such as GPS, GLONASS, and Galileo in time-frequency measurement, time-synchronization detection, space-borne atomic frequency standard comparison, phase noise measurement, etc. [20], therefore, the proposed method has the same applicability in the frequency and phase synchronization measurement of the GNSS system. So it is widely used in radar, satellite positioning, astrometry, communication, precise navigation, aerospace, satellite launch, power transmission, and other high-tech fields.

## Figures and Tables

**Figure 1 sensors-24-04606-f001:**
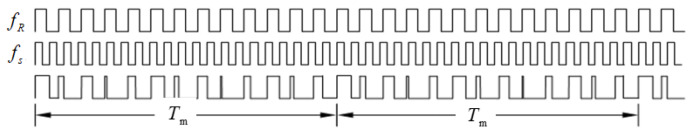
Formed the least common multiple period by different frequency comparisons.

**Figure 2 sensors-24-04606-f002:**
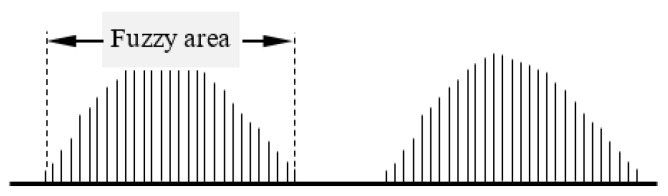
A fuzzy are formed by phase coincidences.

**Figure 3 sensors-24-04606-f003:**
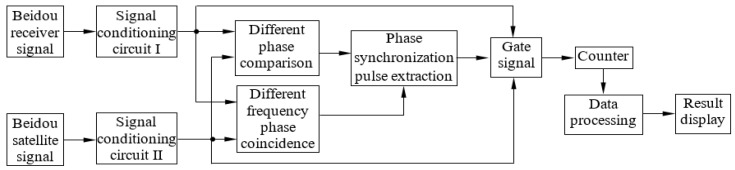
Design a different-frequency phase synchronization detection scheme.

**Figure 4 sensors-24-04606-f004:**
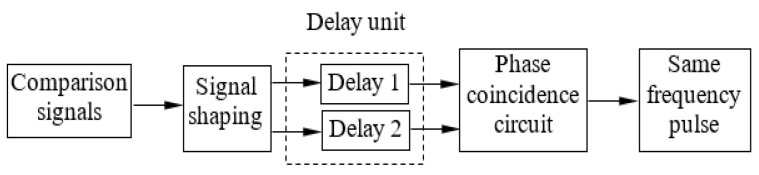
Design the same frequency pulse circuit to improve measurement precision.

**Figure 5 sensors-24-04606-f005:**
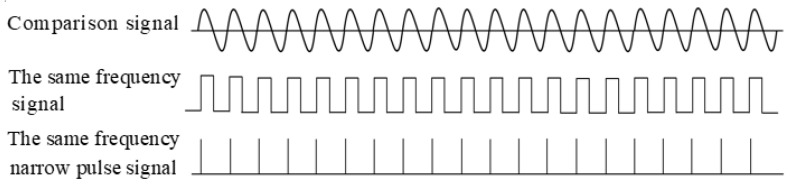
Produce experiment results using the same frequency pulse circuit.

**Figure 6 sensors-24-04606-f006:**
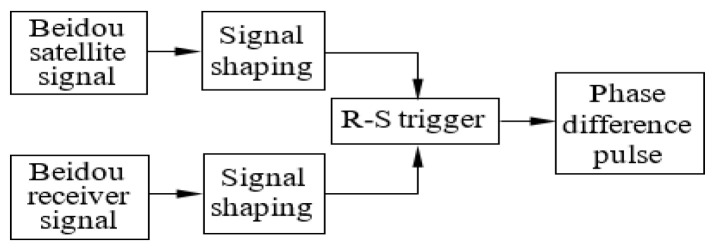
Measure different frequency phase differences.

**Figure 7 sensors-24-04606-f007:**
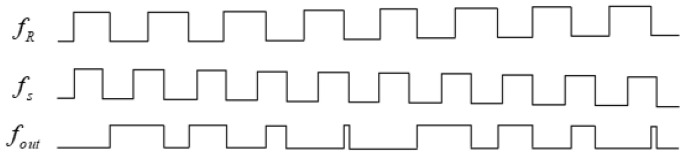
Generate different frequency-phase comparison experiment results by using different phase comparators.

**Figure 8 sensors-24-04606-f008:**
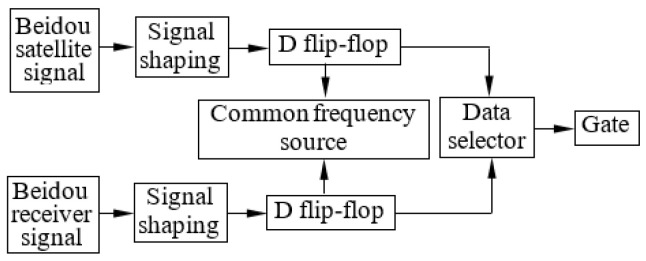
Design a gate switch signal generation circuit.

**Figure 9 sensors-24-04606-f009:**
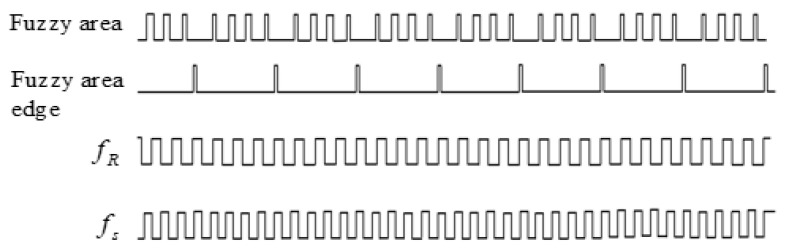
Produce phase synchronization results.

**Figure 10 sensors-24-04606-f010:**
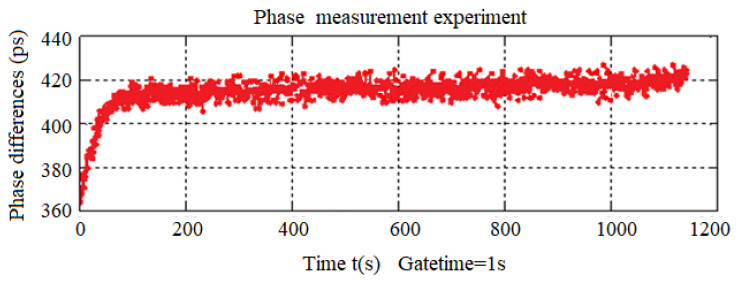
Generated phase difference measurement experiment results by the scheme. The red part of the figure represents the phase fluctuation caused by noise.

**Table 1 sensors-24-04606-t001:** Frequency measurement experiment results.

Measured Frequency (MHz)	Measuring Results (Hz)	Frequency Difference (Hz)	Frequency Stability (s^−1^)
6.4960	6,495,999.455	0.544	5.26 × 10^−13^
10.354	10,353,999.188	0.811	2.35 × 10^−13^
12.885	12,884,999.261	0.738	9.91 × 10^−13^
16.384	16,383,999.849	0.150	1.75 × 10^−13^
18.697	18,696,999.848	0.151	3.65 × 10^−13^
10.000	9,999,999.884	0.115	1.54 × 10^−14^

## Data Availability

Data are contained within the article.

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
