# Peer review of "High-Accuracy Phase Frequency Detection Technology Based on BDS Time and Frequency Signals"

_sensors, 2024, doi:10.3390/s24144606_

Round 1

Reviewer 1 Report

Comments and Suggestions for Authors

The manuscript proposed a phase frequency measurement method for BDS signals based on phase group synchronization. The process is innovative and insightful, reflecting the author's profound professional skills. However, before this article is published, several key issues must be resolved first.

1.      The background and research status of the main research content of this paper need to be further enriched, especially the relevant content of BDS time-frequency signal measurement in recent years.

2.      In Section 2, Line 74 – Line 81, the paragraph states that the frequency deviation between the satellite and the receiver is a certain quantity, which is composed of the greatest common factor, etc... This opinion is confusing. Due to the influence of various error sources in signal propagation and the Doppler effect caused by the relative motion between the satellite and the receiver, the frequency of the satellite signal received by the receiver is not standard and is time-varying. Can you provide a reference for this view, or provide a detailed theoretical analysis?

3.      The paper describes the proposed method in great detail but does not consider the frequency-varying characteristics of the received satellite signal. Therefore, the feasibility of the process in practical application may be questioned. The experiment used in the manuscript to verify the method is a simulation experiment. Compared with the actual propagation of satellite time-frequency signals, the simulation conditions are excessively simple. In order to improve the credibility of the experimental verification, can you provide actual measurement data using a GNSS receiver to collect real BDS satellite signals for the phase measurement example?

4.      The biggest problem is in Section 5, Line 435. It is said that “… BeiDou Doppler frequency shift measurement system is developed”.  However, the previous article did not mention the solution to Doppler frequency shift, which should be the primary consideration of this paper. This conclusion is unfounded.

Comments on the Quality of English Language

The language expression of the manuscript needs further revision. There are some word errors and tense errors. Some sentences are not smooth and difficult to read.

Author Response

1.The background and research status of the main research content of this paper need to be further enriched, especially the relevant content of BDS time-frequency signal measurement in recent years.

Respone 1: Agree. Thank you for pointing this out. We have revised the introduction part of the paper and added the latest references of this year.

2.In Section 2, Line 74 – Line 81, the paragraph states that the frequency deviation between the satellite and the receiver is a certain quantity, which is composed of the greatest common factor, etc... This opinion is confusing. Due to the influence of various error sources in signal propagation and the Doppler effect caused by the relative motion between the satellite and the receiver, the frequency of the satellite signal received by the receiver is not standard and is time-varying. Can you provide a reference for this view, or provide a detailed theoretical analysis?

Respone 2: Agree. Thank you for pointing this out. We have provided two references for this view, such as the reference [16] and [17].

3.The paper describes the proposed method in great detail but does not consider the frequency-varying characteristics of the received satellite signal. Therefore, the feasibility of the process in practical application may be questioned. The experiment used in the manuscript to verify the method is a simulation experiment. Compared with the actual propagation of satellite time-frequency signals, the simulation conditions are excessively simple. In order to improve the credibility of the experimental verification, can you provide actual measurement data using a GNSS receiver to collect real BDS satellite signals for the phase measurement example?

Respone 3: Agree. Thank you for pointing this out. Of course, this is done for the best. However, even for the actual Beidou satellite signal, when the satellite-ground phase frequency synchronous measurement is carried out, the satellite signal should also be down-converted frequency, which is consistent with the actual measurement experiment conducted by us. Considering the cost and the convenience and flexibility of the research, we feel whether it is necessary. However, according to the expert's warning, we are preparing to do this.

4.The biggest problem is in Section 5, Line 435. It is said that “… BeiDou Doppler frequency shift measurement system is developed”. However, the previous article did not mention the solution to Doppler frequency shift, which should be the primary consideration of this paper. This conclusion is unfounded.

Respone 4: Agree. Thank you for pointing this out. Thank you again for your questions. Yes, we also felt that it was unwarranted and inappropriate to draw such a conclusion, so we decided to delete these useless arguments.

Reviewer 2 Report

Comments and Suggestions for Authors

This article introduces an innovative frequency measurement method that is unique and suitable for publication. Although there are some minor issues, only minor modifications are required, and no further review is necessary.

Author Response

1.This article introduces an innovative frequency measurement method that is unique and suitable for publication. Although there are some minor issues, only minor modifications are required, and no further review is necessary.

Respone 1: Agree. Thank you for pointing this out. We have made the revision according to the expert's opinion.

Reviewer 3 Report

Comments and Suggestions for Authors

In this manuscript, the authors theoretically and experimentally discuss the implementation of a phase/frequency measurement technique dealing with different frequency signals. The scenario of interest is BeiDou GNSS constellation. Although the topic is very interesting, I think the authors should carefully revise the paper, since it seems to the reviewer that less attention has been given to a correct writing of the paper (in some cases, for instance, the reference are written as “text”, and not as the common notation of having numbers, while in some other parts of the paper, they are correctly written).

Moreover, the mathematical part is very difficult to follow, please fix the font and try to compact the notation.

I suggest also the authors to consider putting a reference to real-time hardware phase estimators, such as the ones employed in phase interferometry when performing Angle-of-Arrival (AoA) estimation (e.g. reconfigurable digital architectures for AoA estimation which operate in real-time). Another reading I suggest to the authors involves the classical PFD implementation techniques, which are in the paper “Design of Monolithic Phase-Locked Loops and Clock Recovery Circuits - A Tutorial” by B. Razavi, 1996.

Last, I suggest to carefully perform an English revision, and to make sure al acronyms have been declared before using them.

Comments on the Quality of English Language

I suggest to carefully perform an English revision, and to make sure al acronyms have been declared before using them.

Author Response

1. In this manuscript, the authors theoretically and experimentally discuss the implementation of a phase/frequency measurement technique dealing with different frequency signals. The scenario of interest is BeiDou GNSS constellation. Although the topic is very interesting, I think the authors should carefully revise the paper, since it seems to the reviewer that less attention has been given to a correct writing of the paper (in some cases, for instance, the reference are written as “text”, and not as the common notation of having numbers, while in some other parts of the paper, they are correctly written).

Respone 1: Agree. Thank you for pointing this out. We have modified the references, and standardized annotations.

2. Moreover, the mathematical part is very difficult to follow, please fix the font and try to compact the notation.

Respone 2: Agree. Thank you for pointing this out. We have simplified some of the mathematical notation to make it easier for the reader to understand.

3. I suggest also the authors to consider putting a reference to real-time hardware phase estimators, such as the ones employed in phase interferometry when performing Angle-of-Arrival (AoA) estimation (e.g. reconfigurable digital architectures for AoA estimation which operate in real-time). Another reading I suggest to the authors involves the classical PFD implementation techniques, which are in the paper “Design of Monolithic Phase-Locked Loops and Clock Recovery Circuits - A Tutorial” by B. Razavi, 1996.

Respone 3: Agree. We have added two references, such as reference [14] and [15].

4.  Last, I suggest to carefully perform an English revision, and to make sure all acronyms have been declared before using them.

Respone 4: Agree. All abbreviations are declared before application, such as MCU, BDS, LCD and so on.

Round 2

Reviewer 1 Report

Comments and Suggestions for Authors

As mentioned in the last revision, you claimed that “when the satellite-ground phase frequency synchronous measurement is carried out, the satellite signal should also be down-converted frequency, which is consistent with the actual measurement experiment conducted by us.

It is necessary to make relevant explanations in the experimental conditions, otherwise, many readers will have doubts because the topic of the article is about the BDS time-frequency signal, but the experiment is not.

Comments on the Quality of English Language

In the abstract, line 16, "Therefore, it is widely used in satellite 16 positioning,..."

 It should be "Therefore, it would be widely used in ..."

There are many English expression problems such as this. The authors must carefully read the whole text and refine the sentences.

Author Response

  1. As mentioned in the last revision, you claimed that “when the satellite-ground phase frequency synchronous measurement is carried out, the satellite signal should also be down-converted frequency, which is consistent with the actual measurement experiment conducted by us.” It is necessary to make relevant explanations in the experimental conditions, otherwise, many readers will have doubts because the topic of the article is about the BDS time-frequency signal, but the experiment is not.

Respone 1: Agree. Thank you for pointing this out.

In 2019, we also published an article such as Reference [16], High-precision synchronization detection method for bistatic rada, at that time we did not directly use bistatic radar signals, because there is no such equipment in China or even the world at present. We can only do the experiment with down- conversion frequency. This is at least closer to the reality than those methods using MatLab simulation. According to the expert's warning, we are preparing to do this. But we will explain this in the article experiment. Thanks again for the expert's advice, and we will certainly try to do so.

In addition, if our reply is not satisfactory to the experts, we are sorry to say that we have to turn to other publications. Assistant Editor, Ms. Eloise Zhou, has reminded us that the publication fee of 2600CHF is not a small amount when combined into RMB in China. Let's confirm it by email.

  1. In the abstract, line 16, "Therefore, it is widely used in satellite 16 positioning,..."

It should be "Therefore, it would be widely used in ..."

There are many English expression problems such as this. The authors must carefully read the whole text and refine the sentences.

Respone 2: Agree. Thank you for pointing this out. As for the whole paper, we have made English revisions according to the advice of experts.

Reviewer 3 Report

Comments and Suggestions for Authors

My previous comments have been generally addressed. I invite the authors to revise the acronyms notation by writing the extended version when first using a particular word and then the acronym. 

For instance, l. 49: Microcontroller Unit (MCU).

Comments on the Quality of English Language

Minor english editing required. I suggest the authors to carefully check the paper to avoid the presence of any typo.

Author Response

Minor english editing required. I suggest the authors to carefully check the paper to avoid the presence of any typo.

Respone 1: Agree. Thank you for pointing this out. We have carefully check the paper again.